# Perspectives of Molecularly Imprinted Polymer-Based Drug Delivery Systems in Ocular Therapy

**DOI:** 10.3390/polym13213649

**Published:** 2021-10-23

**Authors:** Andreea E. Bodoki, Bogdan-C. Iacob, Elena Dinte, Oliviu Vostinaru, Ovidiu Samoila, Ede Bodoki

**Affiliations:** 1Department of General and Inorganic Chemistry, “Iuliu Hatieganu” University of Medicine and Pharmacy, 400010 Cluj-Napoca, Romania; abota@umfcluj.ro; 2Department of Analytical Chemistry, “Iuliu Hatieganu” University of Medicine & Pharmacy, 400349 Cluj-Napoca, Romania; iacob.cezar@umfcluj.ro; 3Department of Pharmaceutical Technology and Biopharmaceutics, “Iuliu Hatieganu” University of Medicine & Pharmacy, 400012 Cluj-Napoca, Romania; edinte@gmail.com; 4Department of Pharmacology, Physiology and Physiopathology, “Iuliu Hatieganu” University of Medicine & Pharmacy, 400349 Cluj-Napoca, Romania; oliviu_vostinaru@yahoo.com; 5Ophthalmology Clinic Cluj, “Iuliu Hatieganu” University of Medicine & Pharmacy, 400006 Cluj-Napoca, Romania; iovidius@yahoo.com

**Keywords:** molecularly imprinted polymers, drug delivery systems, controlled release, eye diseases, functional polymers

## Abstract

Although the human eye is an easily accessible sensory organ, it remains a challenge for drug administration due to the presence of several anatomical and physiological barriers which limit the access of drugs to its internal structures. Molecular imprinting technology may be considered the avant-garde approach in advanced drug delivery applications and, in particular, in ocular therapy. In fact, molecularly imprinted polymers hold the promise to compensate for the current shortcomings of the available arsenal of drug delivery systems intended for ocular therapy. The present manuscript aims to review the recent advances, the current challenges and most importantly to raise awareness on the underexplored potential and future perspectives of molecularly imprinted polymer-based drug delivery systems intended for the treatment of eye diseases.

## 1. Introduction

Vision is responsible for most of the interactions of the human body with the outside world. A function of this importance is performed through complex, yet very fragile peripheral structures, the eyes. In the last decades, the rising life expectancy has favored a demographic transition towards a growing proportion of people over the age of 50, which are particularly at risk of developing non-communicable eye diseases capable of causing vision impairment or even blindness [1]. According to the World Health Organization data from 2019, out of the total population of the world, about 2.2 billion people have visual impairment [2], more than 200 million patients aged 50 and older being diagnosed with moderate and severe visual impairment, making the treatment of eye diseases an important public health problem with significant socio-economic costs [3]. Early detection and adequate medical intervention are necessary to reduce the negative impact of eye diseases on the global health status.

The architecture of the optical system and the eye itself is of the highest complexity. Although an easily accessible sensory organ, it remains a challenge for drug administration due to the presence of anatomical and physiological barriers which may limit the access of drugs to the internal eye structures [4]. Considering the physiological limitations of an efficient and non-invasive drug delivery to distinct segments of the eye, aligned with the binding requirements warranting patient compliance, there is still a pressing need of going beyond state-of-the-art in the field of ocular drug delivery systems (DDS). Research performed in recent years have primarily focused on the optimization of conventional formulations aiming to boost the pharmacological potential of the currently used drug molecules. 

Giving new valences to already established polymeric materials coming from various fields of science and technology may represent a fast-track solution to further develop intelligent platforms acting as advanced drug reservoirs and delivery systems. The features of molecularly imprinted polymers (MIPs) demonstrated in separation science, align and compensate for the current shortcomings of the available arsenal of DDS intended for ocular therapy, such as limited loading capacity, suboptimal release time and profile, all converging towards an insufficient bioavailability of the active pharmaceutical ingredient (API).

The present manuscript aims to review the recent advances, the current challenges and most importantly to raise awareness on the underexplored potential and future perspectives of molecularly imprinted polymer-based drug delivery systems intended for the treatment of eye diseases.

## 2. Eye’s Structure and Therapeutical Approaches for Its Treatment

The eye is one of the few areas of the body with immune privilege. The eye attempts to limit the local immune and inflammatory response to preserve transparency, and in the end, vision. Multiple mechanisms combine to provide this privilege: physical barriers (corneal epithelium, blood-retinal barrier, and retinal pigment epithelium), soluble factors that inhibit the activity of immune-competent cells (including TGF-β and neuropeptides), and anterior chamber immune deviation (ACAID), a unique response to antigens entering the eye, also a mechanism of tolerance to tissue-specific antigens in the healthy eye [5].

The very surface of the eye, the cornea, must carry multiple functions and is the target of many of the DDS applied topically. It must remain transparent to function as a high-power lens (about 44 dioptries), but also must retain integrity as various aggressions constantly bombard it. Corneal surface is designed to be a highly effective barrier, restricting the access to the eye of high-volume molecules, including pathogens. The cornea, together with the lacrimal film, the conjunctiva, and the eyelids, form a functional system that maintain a double homeostasis of the ocular surface–tissular and optical.

Cornea has a thickness of about 550 microns. It consists of three layers with different characteristics: epithelium, stroma, and endothelium, separated by Bowman and Descemet membranes. Epithelial cells have tight junctions (zonula ocludens), especially in the superficial layers, which provide water-tight seal and strong barrier effect. Corneal epithelium is the main obstacle for drug penetration into the eye [6]. Stroma is the thickest part of the cornea and consist of collagen fibrils organized in lamellae, with small number of cells. The endothelial layer, the innermost of the cornea, functions as an ionic pump, dehydrating the stroma to about 60% of water. Hydration of stroma or strong immune response would cause loss of transparency and of vision [7]. Bowman (between epithelium and stroma) and Descemet (between stroma and endothelium) membranes separate the corneal layers and have little importance to permeation. Figure 1 presents a section through cornea and its relationship with conjunctiva and sclera, as viewed In vivo with ocular coherence technology.

The corneal route is the main access for drugs deposited onto ocular surface, into the aqueous humor. Conjunctival surface is much greater, and the permeability is better than in cornea. Sclera lying beneath the conjunctiva (Figure 1) is also porous. The conjunctival and scleral permeability is 15 to 25 times better than cornea, and molecular size has lower importance [8]. However, the choroid and the RPE limit the bioavailability of the drug, as being the case with subconjunctival injection approach. 

The corneal epithelium is lipophilic and limit the permeation of hydrophilic molecules. Intercellular pores measure 2.0 ± 0.2 nm in diameter that allow only drugs with a molecular weight of <500 Da/10 Å to penetrate [9]. The stroma, on the other hand, acts as a barrier for lipophilic compounds. The endothelium is more permeable than the other two layers, especially for hydrophilic compounds. 

The physiological barriers against the APIs and the need to increase the concentration of drugs in the inner structures of the eye have been permanent challenges for researchers in the attempt to optimize conventional pharmaceutical formulations but also to design new DDSs and to approach alternative routes of administration. 

An ideal ocular DDS ought to be fully biocompatible, to provide a controlled release for extended periods of time, to be compliant with non-invasive routes of administration and last but not least, to be in agreement with a patient-friendly posology. 

The majority of eye pathologies are addressed by topical administration, most often in the form of eye drops. A topical drug administration to the eye is convenient and well-accepted by patients, lacking the discomfort associated with injectable administrations, and is preferred in the diseases of the anterior segment of the eye like conjunctivitis, diseases of the iris or glaucoma [10]. Eye drops, as molecular (true solutions), mechanical (suspensions, emulsions) or colloidal dispersions are non-invasive, fast action and cost-effective. Nevertheless, significant challenges lay ahead of an efficient ophthalmic treatment for a topically applied drug, since, in general, less than 5% of the substance applied as eye drops is retained on the ocular surface and attempts the entrance through corneal barrier [11]. Drug formulations applied on the corneal surface, designed to reach the anterior segment of the eye are hampered by a series of precorneal factors like blinking reflex, tear turnover and lacrimation which contribute to a low ocular bioavailability [12]. The fast clearance of drugs applied on the cornea also results in a short duration of the pharmacological effect, often requiring a frequent dosing which may reduce in some cases patient acceptance of the administered treatment. Additionally, a variable proportion of the topically administered drug could enter in the systemic circulation through nasolacrimal duct drainage, with possible systemic adverse reactions. Moreover, the multi-layered cornea itself is a barrier against the access of both hydrophilic and lipophilic drugs to the anterior eye segment [13]. Also, the presence of efflux transport systems in the epithelial cells of the cornea can alter the trans-corneal penetration capacity of certain drugs like anti-glaucoma or antiviral agents [14]. 

Increasing the remanence of the API at conjunctival level ensures an increased absorption and improved tolerance of APIs; this goal can be achieved by using viscosity-modifying agents, the preparation of bioadhesive formulations, or the preparation of gel-forming solutions containing stimuli-responsive excipients [15,16,17].

The use of emulsions as vehicles for water-insoluble drugs has shown increased internalization of the API dissolved in the non-polar phase. The additional use of cationic polymers such as chitosan, able to increase remanence by an interaction with the negatively charged mucin, further improves the bioavailability parameters [16,18]. By gradual dissolution of the dispersed phase, suspensions with optimal particle size ensure high concentrations of sparingly soluble APIs at conjunctival level. The use of excipients that optimize the physical properties of suspensions such as xanthan gum [19] or ion exchange resins [16], which release the API in the presence of tears, is a strategy that improves the API‘s absorption and increases the duration of the therapeutic effect.

Several nanocarriers intended for both the anterior and posterior segment of the eye such as nanomicelles, dendrimers, nanoparticles, nanosuspensions and liposomes showed promising results as means to control drug release and to improve the permeability, the stability, the biocompatibility, and the bioavailability of APIs [17,20]. Liposomes which can efficiently encapsulate both lipophilic, and hydrophilic APIs exhibit structural similarities to biological membranes, good biocompatibility, and were proven to efficiently release APIs in the anterior and posterior segments of the eye [19]. Nanoparticles contain excipients well tolerated by the eye tissues, such as lipids, proteins, natural or synthetic polymers; their surface characteristics and size may favor the internalization kinetics of APIs into the eye [17]. These drug-laden nano-carriers can be used both as fluid colloidal dispersions, or can be incorporated into other pharmaceutical dosage forms, such as semisolid hydrogels or contact lenses. 

Contact lenses are hydrogel-based medical devices that are applied directly onto the cornea, initially designed to correct ametropia. Therapeutic contact lenses (CLs) obtained by the inclusion of APIs in the lens matrix are considered an excellent alternative for the treatment of ocular diseases, particularly for chronic ones. Direct contact with the cornea can ensure prolonged and/or controlled drug release, with numerous studies showing a more then 50% increase in bioavailability as compared to eye drops formulations [21]. The critical quality parameters in the development of therapeutic contact lenses are represented by the convenience in use, a visual comfort, and a good tolerance. 

CL are composed of polymeric hydrogels or silicone hydrogels. Polyhydroxyethyl methacrylate (pHEMA), was the first monomer used to prepare CLs in the 1960s, and, along with N-vinylpyrrolidone (NVP), it remains the monomer of choice for the manufacture of CLs. Both materials are well tolerated as they confer the lens increased hydrophilicity, adequate mechanical properties, water swelling and oxygen permeability. To optimize these characteristics, other monomers or functionalized compounds can be incorporated into the lens matrix [10,22]. 

Intraocular injections are usually employed for an efficient drug delivery to the posterior eye segment in diseases like macular degeneration or diabetic retinopathy. Intravitreal injections were developed to bypass the natural barriers of the eye but at greater cost, inconvenience, and risk for the eye: infections, haemorrhages, rise of intraocular pressure, or unwanted perforations of the retina. Moreover, the distribution of the drugs in the vitreous body is variable, subjected to additional pathophysiological influences, and the administration could be painful, affecting patient acceptance [23]. Intravitreal injections are commonly used to deliver anti-VEGF drugs (bevacizumab, ranibizumab, aflibercept) inside the vitreous cavity. Due to their protein structure (being either monoclonal antibodies or soluble receptors) these drugs have long intravitreal half-lives but still require a monthly or bi-monthly administration, sometimes for many years, to control the aggressive form of AMD, the neovascular form [24]. Another example are antibiotics which are rarely effective for intraocular infections (endophthalmitis) upon topical or systemic administration, including intravenous, and must be delivered through intraocular injection. 

Implants are devices designed to ensure a prolonged effect of the drugs in the eye, while avoiding frequent intraocular injections, especially in the case of chronic vitreo-retinal diseases. Administration of biodegradable implants (usually PLGA- or polycaprone- based) are preferred due to their high tolerance [17]. Intravitreal administration of a biodegradable PLGA-based implant, for example, was proven to ensure the prolonged release of dexamethasone for 6 months, with beneficial effects in reducing the risk of vision loss in patients with macular edema [17]. 

Another treatment strategy proposes the use of microneedles technology, that can ensure the controlled release of high concentrations of APIs in the retina/choroid. The approach is promising for eye diseases that threaten vision such as macular degeneration, diabetic retinopathy and posterior uveitis [17].

Alternative periocular routes like retrobulbar, peribulbar, sub-Tenon’s or subconjunctival injections are less invasive, being also capable to deliver drugs to the posterior segment of the eye by transscleral diffusion or through choroidal plexus, but usually in suboptimal concentrations due to the presence of blood-ocular barriers which can limit drug access [25].

Generally, a systemic administration of drugs is rarely used in ophthalmic diseases due to a low intraocular bioavailability of the active substance. Despite the easy access to the eye via the vascular choroid plexus, a systemically administered drug must pass through the anterior blood-aqueous barrier (BAB) or the posterior blood-retinal barrier (BRB) in order to gain access inside different segments of the eye. Particularly, tightly packed retinal pigment epithelial (RPE) cells in the outer blood-retinal barrier can significantly reduce the access of a systemically administered drug to the retina and vitreous body, usually less than 2% of the administered dose being available [26]. Additionally, a possible pre-systemic metabolism of oral drugs or the apparition of significant adverse reactions may furthermore limit the systemic administration to a few clinical emergencies such as the acute angle-closure glaucoma [27].

## 3. Drug Loading Strategies in Polymeric Matrices for Extended Drug Delivery

Biocompatible polymeric matrices are convenient scaffold materials for the development of drug reservoirs intended for the prevention and treatment of various eye conditions. To increase the API’s bioavailability and the duration of its therapeutic effect, several approaches for drug loading in polymeric matrices may be used: soaking (including supercritical fluid technology), chemical functionalization of the polymeric matrix aiming to induce guest-host interactions (e.g., cyclodextrins), polymerization in the presence of ionic compounds to promote electrostatic interactions with the API, carrier-mediated release (the incorporation of API in nanoparticulate colloidal systems, such as liposomes, micelles, microemulsions and polymer-based nanoparticles, followed by the loading of these systems into the polymeric scaffold), and molecular imprinting [20,22].

### 3.1. Hydrogels

Hydrogels, mainly designed for topical application (i.e. conjunctival sac), represent networks of natural or synthetic monomers, crosslinked with multifunctional linkers, forming flexible structures upon free-radical polymerization initiated thermally or photochemically. These polymers are insoluble in water but must exhibit a swelling behavior in aqueous solutions while preserving the imprinted sites. Matrix swelling followed by the API‘s release at a certain rate represent fundamental properties of hydrogels and are influenced by the nature of the hydrophilic functional groups and by the degree of crosslinking. The crosslinking of the polymer can be achieved physically or chemically.

Hydrogels can be specifically designed to obtain predictable characteristics (SMART hydrogels) by the appropriate choice of crosslinkers and by the changing of the crosslinking degree. Thus, it is possible to modulate the release rate of the API and the duration of the therapeutic effect by modeling the physical and mechanical characteristics of the hydrogel (such as viscosity, porosity, swelling, mechanical strength, and erosion rates, etc.). Moreover, the addition of functional crosslinking agents enables the gel to react to external stimuli, with major implications on the designed formulation’s pharmacokinetics. Consequently, in situ gel formation at body temperature may occur, or the API‘s release may be triggered by changes in pH or photo-stimulation. 

Natural monomers used in the design of hydrogels include agar, alginic acid, collagen, chitosan, gelatin, fibrin, hyaluronic acid (HA), pectin, etc. The resulting polymers are biocompatible and biodegradable as they share a similar structure to ocular tissues (vitreous humor), and have been intensively studied for tissue engineering. Synthetic polymers, such as polyvinylpyrrolidone (PVP), poly(acrylic acid) (PAA), polyacrylamide (PAM), poly(vinyl alcohol) (PVA), poly(hydroxyethyl methacrylate) (PHEMA), poly(ethylene oxide) (PEO), and cellulose derivatives have lower biocompatibility; however, their manufacturing characteristics are more reproducible. 

Hydrogel forming polymers are currently used in practice for the design of a wide variety of pharmaceutical formulations in ocular drug delivery such as gel-forming solutions, intravitreal hydrogels, topical hydrogels, and contact lenses.

### 3.2. Molecularly Imprinted Polymer (MIP)-Based as Drug Reservoirs

Similar to hydrogel synthesis but exploiting a more tailored interaction between the API and the selected functional monomers, both chemically and sterically complementary binding sites (drug-specific cavities) towards the API molecule may be formed in a highly crosslinked polymeric network, through the process of molecular imprinting [28]. The formation of such specific binding sites within the polymeric network enables a significantly higher drug loading, as well as a considerably extended and more controlled drug release in comparison to the aforementioned hydrogels. The beneficial consequences envisaged upon their clinical application lay in the increase of therapeutic efficiency while maintaining non-invasive routes of administration, in an improved patient compliance especially in chronic eye conditions, and last but not least, in a notable cutback of the amount of drug needed to successfully treat ocular diseases.

MIPs can be rationally designed to obtain predictable characteristics by the appropriate choice of functional monomers and crosslinkers, and their molar ratio with respect to the API. Since their inception, MIPs were successfully developed and applied in analytical techniques, therefore the vast majority of MIP-based DDSs later developed were made from the same acrylic monomers and employing the non-covalent imprinting approach. The most adopted functional monomer is meth(acrylic) acid since its carboxyl group can form weak electrostatic attractions and/or functions as a hydrogen donor for H-bond acceptor drugs; most often it is coupled with EGDMA, as crosslinker. Furthermore, in ocular therapy, the traditional CLs were also based on acrylic monomers (e.g., HEMA), thus the main adaptation of the polymerization mixture formulation was the decrease in the crosslinker concentration. Even though these polymers are already acknowledged as being biocompatible, the development of MIP-DDS for drug delivery applications is still in its infancy. 

In advanced ocular therapy, contact lenses acting as drug reservoirs are able to provide a sustained or controlled release over the wear duration. As formerly mentioned, there are two strategies for the incorporation of the API into the hydrogel: (i) post-synthesis, via the traditional approach of soaking the premade lens in a drug solution or (ii) during hydrogel polymerization. The first approach offers the advantage of simplicity and ease of fabrication, but it provides a drug release profile characterized by an initial burst release with potentially serious consequences for the patient, followed by subtherapeutic concentrations. In the latter case, drug incorporation occurs simultaneously with the lens synthesis process. Here a distinction needs to be made between the entrapment of drug during polymerization and the molecular imprinting approach. Not every polymer made in the presence of a drug molecule will result in a MIP, because the imprinting effect is not determined by the mere presence of a template in the polymerization mixture. The success of the imprinting depends upon the effectiveness in achieving imprinted cavities in a crosslinked polymer matrix, characterized by (i) shape complementarity to the template molecule (i.e. targeted drug molecule, API) and (ii) specific spatial distribution of the functional groups able to promote multiple, mostly chemical interactions, with the template. This can only be achieved by optimizing the imprinting parameters for each API, since the template and its functionalities determines the choice of the functional monomer(s). As a matter of fact, a complex combination of factors including the nature, concentration and the ratio of the functional and crosslinking monomers, the functional monomer-template ratio, polymerization conditions (temperature, UV, duration, etc.) strongly affect the process of molecular imprinting [28]. Following this optimization, the stability, and the number of the imprinted cavities, as well as the MIP’s affinity towards the template can be adjusted, which in turn dictates the amount of drug loaded and the control over its release profile. In principle, the efficiency of the molecular imprinting process is assessed in comparison to a reference, non-imprinted polymer (NIP). 

On the other hand, in cross-linked polymer-based contact lenses in which the API is added during the polymerization process, its molecule is retained in the hydrogel’s polymeric network mainly by physical entrapment. Even though a so called “functional monomer” (i.e. substance capable of interacting with the drug) is used for CL production, they are not able to generate imprinted cavities due to the lack of sufficient and spatially-arranged functionalities that can provide multiple interaction points with the drug. Furthermore, if a subsequent drug washing step is performed, no “drug-specific” cavities will remain in the polymeric matrix, as the lack of effective crosslinking will cause their collapse. For all these reasons, the drug-entrapped contact lenses usually demonstrate low drug loading and a burst release of the API with poor control over the release behavior in comparison with the molecularly imprinted polymers.

## 4. MIP-Based Systems Intended for Ocular Drug Delivery

There is a long and sinuous road from the early days of these functional polymers towards the MIP-based drug delivery systems intended for ocular therapy, with several noteworthy milestones (Figure 2). In developing ocular drug delivery systems various aspects of the conventional molecular imprinting process need to be carefully adapted for the intended application field and preferred therapeutic strategy, but also aligned to the pharmaceutical requirements and regulations.

MIPs have been initially developed for and employed mainly in analytical applications, as stationary phases/selective sorbents in separations sciences [29] and more recently as interfaces in the development of (bio)chemical sensors. The main feature which differentiated them from the traditional stationary phases/interfaces was their predetermined ligand selectivity, induced during the synthesis process. The standard MIP fabrication procedure involves the dissolution in a proper and inert solvent, also called porogen, of the following four components: template (target) molecule, functional and crosslinking monomers and an initiator. After the polymerization step, the template is washed out, leaving behind permanent cavities generated into the three-dimensional polymer matrix, complementary in shape, size, and functionality to the template molecules. The obtained polymers were used for their abilities to selectively rebind the template molecules or its structurally similar analogues. 

However, in the case of MIP-based drug delivery systems, by the nature of their intended use as drug reservoirs with prolonged and, ideally, controlled release, the template removal step seems to be redundant, as long as any potentially unreacted component of the polymerization mixture holds no toxicity for the human tissues. If, however, a thorough wash needs to be carried out (e.g., when monomers with measurable cellular toxicity need to be used), a subsequent drug loading procedure is performed [28]. Obviously, this scenario is objectionable because it will greatly prolong the DDS processing step, may pose additional or unexpected toxicity issues, and it will substantially increase the total amount of drug used. 

In ocular drug delivery, the template-functional monomer interactions are based mainly on weak, non-covalent associations responsible for faster binding and release properties during the analyte-polymer interactions, compared to the covalent imprinting approach. Most of the conventional MIPs intended for analytical applications are synthesized in organic aprotic solvents in order to stabilize the template-functional monomer complex during polymerization, but also to confer a macroporous polymeric structure enabling thus an easier diffusion of the template molecule from and into the imprinted cavities. Nevertheless, the presence of residual organic solvents is undesirable in drug delivery applications and especially in the case of ocular DDSs. Moreover, the use of organic solvents in CLs fabrication is often conflicting and thus liquid monomers need to be exploited for template dissolution. 

As in the case of MIPs intended for analytical applications, the performances of the imprinted hydrogels developed for drug delivery purposes must be compared with the corresponding NIPs, which are prepared analogously as the imprinted ones, but without the addition of the template drug. In analytical applications (separation and sensing), one of the main features of the obtained functional polymers is selectivity, the supreme goal of an analyst being the specific binding of the target analyte. In drug delivery, however, other fundamental features are primarily sought after, such as: drug loading and releasing properties. Although, seemingly aiming for different objectives based on the field of their application, in fact these features are somewhat interconnected, as higher binding affinity of the template to the imprinted cavities generally implies improvements in drug loading capacity and may hold the premises for prolonged/controlled drug release. As being outlined in the previous section, MIP-based DDS may alleviate shortcomings related to the low bioavailability of various eye drops, non-imprinted polymeric hydrogels and ointments intended for non-invasive, topical, and especially long-term administration, stemming from their high physiological clearance, limited drug-loading, or excessively fast drug release rates, thus holding the promise of a fascinating area of research. Judging by the number of published research in the last two decades (Table 1), the topic of molecularly imprinted DDS for ocular therapy seems to be severely underexplored, for grounds that the following sections will attempt to unveil.

### 4.1. Advantages and Limitations of Molecular Imprinting in Ocular DDS Development 

#### 4.1.1. Advantages

Maybe the most promising direction in the current field of topical ophthalmic drug delivery is the molecular imprinting approach as it can offer real solutions to the problems exhibited by the conventional ocular drug delivery formulations (e.g., ophthalmic solutions, ointments, non-imprinted hydrogels, CLs) such as low bioavailability, exceedingly fast release profile, or insufficient loading capacity. Therefore, by creating imprinted binding sites in the polymer network in which numerous functional groups are concentrated and organized in such a manner to attain a strong interaction with the API, a better control over the release profile can be achieved. In fact, using MIPs as drug reservoirs, a sustained, zero-order drug release may be obtained for a long period of time, when compared to conventional drug delivery [28]. The residence time of the drug within the polymer can be adjusted by controlling the number and the strength of interactions between the template and the polymer. This is achieved by opting for the proper imprinting approach (covalent or non-covalent) followed by the empirical or computational modeling-based selection of suitable, but biocompatible, functional monomer(s) available from a large library. 

Furthermore, the tailor-made imprinted pockets of the MIPs may help in protecting the active ingredient upon storage or therapeutic use from enzymatic, hydrolytic or photo-degradation, maintaining the drug for a longer period in its bioactive form, thus increasing its bioavailability. 

The molecular imprinting technique can be applied to any kind of template, ranging from ions to relatively small pharmaceutical compounds and even large (bio)molecules, like proteins. However, the increase in the size of the template, may hamper its efficient release from the cross-linked polymer, risking of being permanently stuck inside. 

From a pharmacological point of view, a very wide variety of APIs were used as templates in the process of imprinted-hydrogel synthesis, such as: β-blockers (timolol as antiglaucoma medication [30,31,32,59]), α2 adrenergic agonist (brimonidine as antiglaucoma medication [52]), antihistamines (ketotifen [36]), non-steroidal anti-inflammatory agents (ibuprofen [49], diclofenac [42]), corticosteroids (prednisolone [48], fluorometholone [50]), antibiotics (ciprofloxacin [37], moxifloxacin [42], minocycline [43], polymyxin B [44], vancomycin [44]), antifungal drugs (voriconazole [45]), antiviral drugs (acyclovir and valacyclovir [46]), carbonic anhydrase inhibitors (dorzolamide [55], acetazolamide [53,54] used as antiglaucoma medication), statins (atorvastatin [56]), bioprotectants (trehalose [49]), hydroxypropyl methylcellulose used to relieve dryness and irritation caused by reduced tear flow [58], hyaluronic acid used in the treatment of dry eye and other conditions [57].

In most cases, various controlled drug release platforms, MIPs included, have focused on the incorporation and the release of a single active molecule, either a comfort molecule or a pharmaceutical agent. However, MIPs, as drug reservoirs, may be adapted for the simultaneous and controlled delivery of multiple active molecules [49]. 

In conventional drug delivery, the API is loaded into the gel after synthesis by equilibrium partitioning which sometimes requires even days, in contrast to the imprinting approach where the gel is synthesized in the presence of the drug and the obtained imprinted polymer is already fully loaded. 

#### 4.1.2. Challenges 

Despite its great potential in drug delivery applications, molecular imprinting bears several concerns with respect to ocular therapy. The most important one is finding the right balance between a rigid cross-linked network offering high-fidelity imprints and an elastic hydrogel displaying good optical and pharmacokinetic properties. The presence of a template molecule in the polymerization mixture may lead to a polymer with different morphology and swelling properties [51,58]. Therefore, in principle, for each API an optimization of the imprinting parameters is required to achieve a biocompatible hydrogel with the proper elasticity, optical and swelling properties. For example, imprinting CLs with bimatoprost showed increased drug loads and better release profiles, but with a negative impact on the morphology and the behavior of the MIP from a drug delivery point-of-view [51]. 

The concentration of the functional monomer, especially if it is endowed with hydrophilic functional groups, influence the swelling and the water content of the resulting hydrogels. By using more functional monomer, which is also beneficial in terms of drug loading capacity, the polymer’s water content will be increased [48]. Several studies showed however that there was no significant difference in the water content between the imprinted and non-imprinted lenses and the molecular imprinting method did not influence the viscoelastic, swelling and transparency properties of the contact lenses [37]. 

### 4.2. Balancing between Flexibility of CLs and Imprinting Efficiency

In the MIP synthesis protocol, one compulsory component of the polymerization mixture is the crosslinker whose role is to provide a reticulated three-dimensional polymer matrix and to preserve the structural integrity of the imprinted cavities. The conventional imprinted polymers designed for analytical applications (i.e. chromatographic adsorbents) are relying on a relatively high degree of cross-linking (50–90% cross-linker agent [60]) in order to achieve imprinted sites with maximum fidelity and a porous polymer matrix ensuring high linear flow velocities of the mobile phase and a rapid mass transfer of the targeted analyte (template).

The imprinted hydrogels, and especially the CLs, need to be flexible enough to ensure good biocompatibility and swelling behavior (water content). A high degree of crosslinking also affects the optical clarity and transparency of CLs [61]. Therefore, from a drug delivery perspective, the composition of CLs should contain low concentrations of crosslinker(s). A maximum of 5–10 mol% crosslinker is generally recommended [61,62]. The most employed crosslinker in the development of imprinted CLs is by far EGDMA, presenting two vinyl groups. Up to now, the emphasis in the published studies was on the cross-linker’s concentration and less on its nature.

Achieving the right balance between high-fidelity imprinted sites and elasticity of the CLs it is not an easy and straightforward process and sometimes requires laborious optimization procedure for each targeted API. One of the most important considerations in developing imprinted CLs for drug delivery purposes is to preserve their biocompatibility and swelling behavior, as the latter is affecting the oxygen dissolution and diffusion into the surface of cornea [63]. Therefore, it is not recommended to sacrifice these features for a modest improvement in the imprinting efficiency.

The nature of the functional monomer is another essential factor affecting the imprinting efficiency. The ability of functional monomers to interact with drug templates during polymerization is different. Thus, the selection of co-monomers significantly influences the performance of the final MIP hydrogel, such as release rate, optical clarity, swelling and mechanical properties [58]. Sometimes, a mixture of functional monomers is applied to obtain an optimized MIP with higher loading capacity and improved release kinetics [35,57].

First studies investigated the influence of the functional monomer:template ratio using the empirical approach, in which the concentration of monomer was kept constant while the concentration of the template was varied [48]. As expected, and in line with the non-covalent imprinting approach, increasing the functional monomer content leads to an increase in the affinity of the polymer and thus in the loading capacity. In return, lower ratios of functional monomer, where efficient template complexation is restricted by the limiting amounts of monomer, favors the release of higher amounts of drug (template) in a given timeframe.

Besides the key role of the functional monomer, functional (charged) additives may also bring unexpected advantages to the efficiency molecular imprinting. Long-chained negatively charged comfort agents, such as hyaluronic acid, commonly used as comfort agent to prevent protein deposition on silicone hydrogels [64], through additional electrostatic interactions with the template (e.g., timolol) may beneficially alter drug loading and its release profile [31,33,59,65]. 

Alternative molecular imprinting strategies, such as metal-ion mediated imprinting, are able to increase the degree of order throughout the polymerization step by well-defined, spatially-directional coordinate bonds [29]. This kosmotropic effect leads to imprinted cavities of higher fidelity, and even to mimic the active sites of enzymes. As efficient ligands of Zn(II), 4-vinylimidazole and 1-vinyl imidazole, resembling aminoacids of the carbonic anhydrase, were used as functional monomers to achieve high-loading molecularly imprinted drug reservoirs for two inhibitors of the metallo-enzyme and in the meantime antiglaucoma drug representatives, namely acetazolamide and its structural analogue, ethoxzolamide [53,54]. Moreover, by playing with the nature of the metal salt (zinc nitrate instead of zinc methacrylate) cytocompatibility and optical transparency of the resulting CLs could also be improved [54].

### 4.3. Computer Modeling

The empirical selection of the polymerization mixture’s components is time and resource-consuming and might not even offer the best performing imprinted hydrogels. The more efficient approach to achieve high affinity MIPs with appropriate morphological and physico-chemical properties is the computer-assisted molecular modeling, by helping in better understand, but also to rank the interactions taking place during the molecular imprinting process. Eroglu et al. [43] conducted an integrated computational (molecular dynamics) and experimental study to assess the relationship between the design parameters of MI and the drug (minocycline) uptake and release performance of the imprinted hydrogels.

The implementation of computer-assisted techniques in the development of imprinted hydrogels can reduce the tedious and time-consuming laboratory work as well as the amounts of used chemicals, which is very important especially in the case of expensive template drugs.

### 4.4. Loading/Release Efficiency

By far the main advantage of MI technology in the design of imprinted hydrogels is overcoming their low affinity for most drugs [62,66]. It was repeatedly demonstrated that generating imprinted sites for a specific drug within the hydrogel, not only gives access to efficiently control the residence time of the drug but can also increase drug-loading capacity.

The imprinting efficiency is determined by the stability of the functional monomer-template complex throughout the polymerization step. Therefore, as expected, the component with the highest impact on the drug-loading capacity, is the nature and the concentration of the functional monomer. Generally, the most favorable monomers for the imprinting process are those with the highest binding affinities to the template [31]. Furthermore, the monomer’s concentration strongly influences drug binding, and up to a critical value being directly correlated with the hydrogel’s loading capacity. Going beyond that level, its beneficial effect may be reversed [31].

Drug loading may depend also on the pH at which the drug-stripped hydrogel is exposed for the re-loading step. For example, HEMA-MAA imprinted hydrogel adsorbs the highest amount of timolol (template), at pH 5.5 and 7.5, at which the carboxylic groups of MAA are ionized and are capable of interacting with the positively charged timolol. At pH values where the non-ionic form of the hydrogel (pH < 5.5) and/or template (pH > 9.2) is predominant, a severe cutback of the drug rebinding occurs [31]. 

The non-covalent interactions between the functional monomer-template are preferred from a drug release perspective due to the faster binding and release kinetics. The delayed kinetics of drug release and slower diffusion rate of the imprinted hydrogels in comparison with the non-imprinted ones, as prerequisites of drug reservoirs designed for long term treatment, are currently explained by the tumbling effect [49], i.e., the migration of the template molecules from one imprinted cavity to another within the hydrogel matrix.

The most notable differences between the release time, but also released amount of drug of the imprinted vs. non-imprinted CLs were observed when exposed to low concentrations of template during drug-rebinding [37]. However, exposed to higher concentrations of template, due to the concurrent increase of non-specific interactions with the polymeric scaffold, drug-release profiles of imprinted and non-imprinted CLs tend to overlap.

As previously mentioned, by the right choice of functional monomer(s) and monomer:template ratio, depending on the therapeutic needs and on the required duration of wear in case of CLs (up to several days, and even longer [49,58]), the rate and duration of drug and comfort molecules’ release can be rigorously programed, creating the premises of simplified treatment plans and improved patient compliance.

Another factor affecting the release behavior is the swelling property of the lens in aqueous media. Upon the rising of water content within the polymer network, the conformation of specific binding sites changes accordingly, which in turn impacts negatively on the hydrogel’s drug binding capacity. The addition of functional monomers (e.g., NVP, DMAA, MAA) to HEMA significantly increases the water content of the hydrogel [62]. However, in the case of hydrophilic drugs, the loading capacity of MIPs with superior swelling properties, is usually higher than that of MIPs with less water content.

Additionally, to avoid potential pitfalls, care must also be taken during In vitro characterization and optimization of the imprinted drug-reservoirs, as the obtained drug-release kinetics may significantly differ for the same formulation in function of the selected experimental models. For example, using an infinite sink dynamic release of ketotifen fumarate from imprinted CLs showed a concentration dependent (Fickian) behavior, whereas testing the same CL under physiological volumetric flow rates and tear volumes using a PDMS microchip, a slower release rate, with an ideal, concentration independent (zero-order) release profile was recorded [35].

### 4.5. Contact Lenses as Promising MIP-Based DDS for Ocular Treatment

A quick survey of the publications (Table 1) shows that to date, CLs are the preferred formulations under study for the imprinted hydrogels as DDSs. Even though the currently marketed vision corrective CLs are widely used clinically today (only in US there are 45 million lens wearers [67]) and the first patent of a drug-releasing contact lens has been around for over 50 years, only a few therapeutic contact lenses are commercially available today. However, with the advent of silicone-based hydrogels which allow an extended wear, the research in this field intensified in recent years. Drug-soaked and entrapped contact lenses have proved that they are not viable options in developing CLs for therapeutic purposes. MI technology turned out to be the avant-garde approach in drug delivery applications and especially for ocular therapy.

The critical features of such imprinted polymer backbones used for CL manufacture are biocompatibility and non-immunogenicity. Due to their topical route of administration, biodegradability it is not a decisive feature as after payload release, they are removed from the surface of the eye. Process control during optimization and manufacturing of CLs should involve the following In vitro tests: optical transparency, water content, wettability, tribology, refractive index, and oxygen and ion permeability. Furthermore, In vivo studies should validate the aforementioned characteristics and should demonstrate (pre)clinical efficiency.

The imprinted hydrogel-based CLs rely mainly on the same acrylic functional (e.g., MAA, MMA, acrylamide, acrylic acid, HEMA) and crosslinking monomers (e.g., EGDMA, TEGDMA) as the ones employed in analytical applications. Therefore, the transfer of this technology in CLs development does not require multiple adjustments, besides lowering the degree of crosslinking.

The other material of choice in mass-fabricating vision corrective CLs is silicone, however only a few studies attempted to employ it in developing silicone-based molecularly imprinted hydrogels [33,37,40,49,58]. The main issues of using silicone in CLs are its hydrophobicity and rigidity causing poor wettability and abrasiveness [68]. Creating co-polymers with hydrophilic monomers (e.g., HEMA) up to now have not lead to significant increase of the bulk water content (~15%) [37], therefore, surface modifications with such hydrophilic comonomers remains the efficient way in increasing water permeability required by the on-eye movement of CL [69]. Nevertheless, it possesses one important advantage, namely the high permeability to oxygen which enables such CLs to be worn for a long period of time. However, increasing the CLs’ water content, their oxygen permeability substantially decreases. 

Commercially available, silicone hydrogel-based contact lens materials (e.g., Lotrafilcon B-containing a mixture of the proprietary silicone hydrogel macromer, TRIS, and dimethyl acrylamide), might not always offer high enough imprinting efficiencies. In this respect, the admix of additional functional monomers (i.e. AA) and crosslinkers (PEG200DMA and EGDMA) lead to considerable enhancement of HPMC loading capacity and release rates [58]. Obviously, the performance of other mixtures of contact lens starting material, containing acrylate or vinyl-pyridine functional monomers were [49] and may be further investigated, as long as the optical properties of the resulting CLs are preserved within acceptable limits.

In the design of therapeutical CLs combined drug loading strategies may also be applied, where the simultaneous and sustained delivery of two or more APIs may be achieved by MI and layer-by-layer (LbL) coating. As such, the inner core of the CL may be imprinted by a drug of interest (e.g., antibiotic, anti-inflammatory agent, etc.), whereas by LbL deposition of various polyelectrolytes (e.g., alginate, poly-L-lysine, sodium hyaluronate), besides the fine-tuning of the drug-release through drug-polyelectrolye interactions, additionally biocompatibility, non-antigenicity, anti-fouling and antibacterial properties may be conferred to the CLs surface [40,70].

To conclude, probably the most promising way of formulating drug-imprinted hydrogels as drug reservoirs is through therapeutic CLs, as it combines the features of a robust drug loading platform, with multiple means of controlling drug release profiles of one or several APIs, while ensuring a patient friendly, non-invasive means of treatment for extended periods of time.

## 5. Challenges and Perspectives of MIPs as Ocular DDS

Even though the ocular bioavailability of drugs applied on the corneal surface through classical ophthalmic preparations (ocular drops and ointments) is very poor (~5%), the use and commercialization of more advanced ocular DDS are very limited, and they seem to have come to a standstill. Therapeutical contact lenses can be a viable alternative as they can enhance ocular bioavailability of drugs, while reducing the side effects and increasing patient compliance. However, the mere soaking and entrapment of APIs into the polymeric matrix of contact lenses do not provide targeted delivery and sufficient control over the release. Despite its early developing stage in the field of drug delivery, molecular imprinting represents the most promising technology to be employed in the development of efficient ocular drug delivery devices. Imprinted contact lenses proved that they have the ability to improve the drug delivery profile, to prolong the release and residence time of the drug and even to favor the drug loading capacity. According to the reviewed studies, good imprinting efficiency can be achieved without sacrificing the critical properties a standard, vision corrective contact lens should possess, such as optical transparency, good mechanical properties, ion and gas permeability, wettability, or refractive index. Nevertheless, occasionally the lack of careful planning of the experimental goals in line with the particularities of ocular therapy, misleading data interpretation may throw off course the development/optimization process (i.e. drug-release kinetics may significantly differ for the same formulation in function of the selected experimental models; immersing a contact lens in the artificial tear solution under sink conditions do not provide realistic results and tend to overrate the release period and behavior).

In spite of the promising results of numerous In vitro studies published in recent years, consistent In vivo data is still missing. Apart from the known limitations brough on by the MI process (non-rational selection of the polymerization mixture’s components, at times, less than ideal optical transparency, gas permeability, etc.), several other possible reasons may be invoked for the scarcity of In vivo reports. In no particular order of importance, one may be the lack of interdisciplinary approach, as the most active research groups dealing with molecular imprinting are still in the field of materials science, with limited access to animal research facilities. Another important reason might be the lack of appropriate animal models (rats, mice) that relevantly translate the observed pharmacokinetic/pharmacodynamic results to human applications. The use of animal models sharing a higher degree of similarity with the human eye (e.g., rabbits, swine, cats, dogs, horses or primates) imply significantly higher costs or more expensive research infrastructure, but most importantly runs into serious ethical issues concerning their inclusion in such studies.

Future studies should focus on expanding the body of knowledge related to the performance of such molecularly imprinted DDSs in order to ensure the fields maturity towards clinical validation which will pave the way for marketing authorization and ultimately improve the patient’s quality of life.

## Figures and Tables

**Figure 1 polymers-13-03649-f001:**
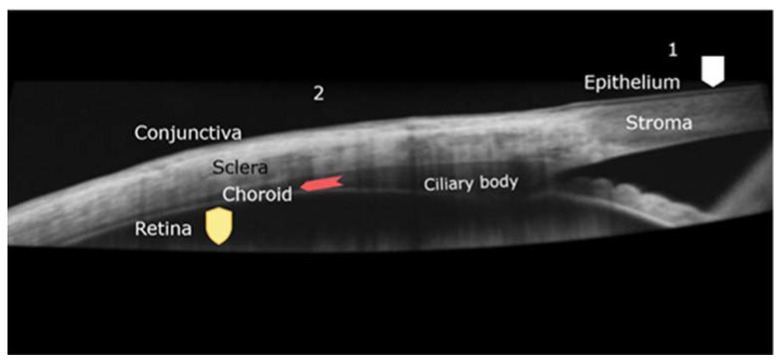
Ocular coherence technology (OCT) image of the junction between cornea and sclera covered by conjunctiva. Beneath the sclera, the choroid and ciliary body are seen. Barriers that prevent drug permeation are highlighted: 1–corneal pathway, epithelium barrier (squared shield). 2-transscleral pathway, choroid circulation (arrow) and retinal barrier (shield).

**Figure 2 polymers-13-03649-f002:**
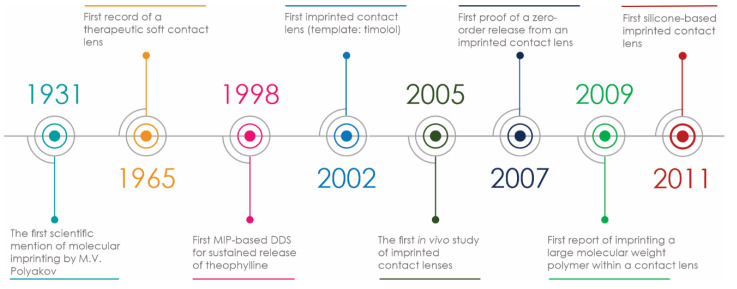
Milestones in the development of MIP-based ocular drug delivery.

**Table 1 polymers-13-03649-t001:** Most relevant studies of the last two decades focusing on the development of MIP-based hydrogels intended for ocular treatment.

Active Pharmaceutical Ingredient	Imprinting Approach Polymerization Mixture (Functional Monomer/CROSSLINKER/INITIATOR)	Polymeric Scaffold Characteristics	Drug Load	Drug Release	Development Stage	Key Observations	References
Timolol maleate	Non-covalent imprinting; MAA/EGDMA/AIBN; 50 °C, 12 h and 70 °C, 24 h	Poly-HEMA hydrogel/CL (700 μm) Water uptake: 108%	12 mg/g hydrogel for MIP vs. 4 mg/g hydrogel for NIP IF = 3	Sustained release (complete release in 12 h; slowest release in artificial lacrimal fluid, pH = 8)	Material characterization In vitro release studies: method–sink model; medium–0.9% NaCl (pH = 5.5), PBS (pH = 7.4), artificial lacrimal fluid (pH = 8), 37 °C		[30]
Timolol maleate	Non-covalent imprinting; MAA/EGDMA/Darocur 1173; UV, 50 min, room temperature	Poly–DEAA hydrogel/CL (300 μm) Water uptake: not reported	IF: approx. 2	Sustained release for 48 h	Material characterization In vitro release studies: method–sink model; medium–0.9% NaCl (pH = 7), 37 °C		[31]
Timolol maleate	Non-covalent imprinting; MAA/EGDMA/Darocur 1173; UV, 20 min, room temperature	Poly–DEAA hydrogel/CL (80 μm) Water uptake: 35–36%	34.7 μg/lens for MIP vs. 21.2 μg/lens for NIP IF = 1.63	*In-vivo* release studies–sustained release: initial pulse release (up to Cmax = 330 μM for MIP vs. Cmax = 183 μM for NIP) followed by exponential decrease	Material characterization In vitro release studies: method–sink model; medium–0.9% NaCl (pH = 7), 37 °C In vivo release studies: Nippon albino rabbits	In vivo release studies: Timolol detected in the tear fluid for 180 min for MIP vs. 90 min and 60 min for NIP and eye drop solution, respectively; MRT-12 min. for MIP and NIP vs. 5.32 min and 6.35 min for 0.068% and 0.25% eye drop solution, respectively	[32]
Timolol maleate	Non-covalent imprinting; -/EGDMA/Irgacure 184; UV, 15 min, room temperature	Poly (HEMA-co-TRIS) and Poly (DMA-co-TRIS) hydrogels/CL (1000 μm) Water uptake: 26.5% and 32.3%	Not reported	Sustained release for 48 h; Combined strategy, MI and addition of hyaluronic acid-highest mass of released drug	Material characterization In vitro release studies: method–sink model; medium–PBS (pH = 7.4), 37 °C	Hyaluronic acid–negatively charged wetting agent used as a functional additive instead of functional monomers to increase loading capacity	[33]
Ketotifen fumarate	Non-covalent imprinting; AA, AM, NVP/PEG200DMA/AIBN; UV, 10 min, 36 °C	Poly-HEMA hydrogel/CL (400 µm and 700 µm) Water uptake: 40–50%	4.9 × 10^−2^ mmol/g hydrogel for MIP	Artificial lacrimal fluid-prolonged release profile for 5 days (2200 µg cumulative release, 80% drug released in 4 days) Lysozyme (1mg/mL) in artificial lacrimal fluid–5-fold increase in the duration of release	Material characterization In vitro release studies: method–sink model; medium–artificial lacrimal fluid (pH = 80), and lysozyme (1mg/mL) in artificial lacrimal fluid	Receptor-inspired hydrogel (microdomains that resemble histamine H_1_-receptors)	[34]
Ketotifen fumarate	Non-covalent imprinting; AA, AM, NVP/PEG200DMA/AIBN; UV, 10 min, 36 °C	Poly-HEMA hydrogel/CL (thickness not reported) Water uptake: not reported	20.85 mg/g hydrogel for MIP	Infinite sink model: Fickian kinetics, prolonged release for 5 days (1200 µg cumulative release, 80% drug released in 4 days) Physiological flow model: Zero-order kinetics, sustained release for 3.5 days (45 µg cumulative release, constant rate of 12.9 µg/day)	Material characterization In vitro release studies: method–sink model, and physiological flow model (microfluidic device); medium–artificial lacrimal fluid (pH = 8)	Zero-order release under physiological volumetric flow rates	[35]
Ketotifen fumarate	Non-covalent imprinting; AA, AM, NVP/PEG200DMA/AIBN; UV, 9 min, 35 °C	Poly-HEMA hydrogel/CL (100 μm) Water uptake: not reported	115 μg/lens for MIP vs. 39 μg/lens for NIP IF = 2.95	In vitro release studies: prolonged release for 72 h (85% of drug released in 24 h for MIP vs. 100% drug released in 6 h for NIP) In vivo release studies: initial pulse release (up to Cmax = 214 μg/mL) followed by a sustained release for MIP vs. initial pulse release (up to Cmax = 140 μg/mL) followed by an exponential decrease for NIP	Material characterization In vitro release studies: method–sink model; medium–artificial lacrimal fluid (pH = 8), 34 °C In vivo release studies: New Zealand white rabbits	In vivo release studies: MRT–12.47 h for MIP, 3.75 and 50 times greater than for NIP and 0.035% eye drops, respectively; sustained, extended release (average constant tear film concentration of 170 μg/mL up to 26 h)	[36]
Ciprofloxacin	Non-covalent imprinting; AA/EGDMA/Irgacure; UV, 20 min and 50 °C overnight	Poly (HEMA-co-TRIS) hydrogel/CL (1000 μm) Water uptake: approx. 15%	Not reported	Sustained release for up to 14 days	Material characterization In vitro release studies: method–sink model; medium–artificial lacrimal fluid (pH = 8)		[37]
Ciprofloxacin	Non-covalent imprinting; AA/EGDMA/AIBN, Irgacure 1173; UV, 5 min, room temperature	Poly (HEMA-co-TRIS-co-NVP) hydrogel/CL (64 μm) Water uptake: 36.2%	1509 μg/lens for MIP vs. 1383 μg/lens for NIP IF = 1.09	Sustained release for up to 8 h	Material characterization In vitro release studies: method–sink model; medium–PBS (pH = 8) In vitro antibacterial activity (*P. aeruginosa*) In vivo testing of antimicrobial activity–Rabbit scratch model (white New Zealand rabbits)	*In-vivo* study: the imprinted SCL have similar performance to conventional antibiotic eye drops	[38]
Ciprofloxacin	Non-covalent imprinting; MAA/EGDMA/AIBN; 50 °C, 24 h	Poly-HEMA hydrogel/CL (400 μm) Water uptake: 34%	170–210 μg/disc for MIP vs. 120–160 μg/disc for NIP IF = 1.31–1.41	Sustained release (60% released in the first 5 h followed by a sustained release profile for 50 h)	Material characterization In vitro release studies: method–sink model; medium–NaCl 0.9%, and artificial lacrimal fluid (pH = 8), 37 °C In vitro antibacterial activity (*P. aeruginosa*, *S. aureus*)		[39]
Moxifloxacin	Non-covalent imprinting; AA/EGDMA/AIBN; 60 °C, 24 h	Poly (HEMA-co-TRIS-co-NVP) hydrogel/CL (300 μm) Water uptake: 130%	64.9 μg/mg hydrogel for MIP vs. 44.1 μg/mg hydrogel for NIP IF = 1.47	Sustained release (60% release in the first 8 h followed by a sustained release of effective concentrations for 13 days; 45 μg/mg hydrogel cumulative release)	Material characterization In vitro release studies: methods–sink model and physiological flow model using a microfluidic device; medium–artificial lacrimal fluid (pH = 8), 36 °C In vitro antibacterial activity (S*. aureus* and *S. epidermidis*)		[40]
Moxifloxacin	Non-covalent imprinting; AA/EGDMA/AIBN; 60 °C, 24 h	Poly (HEMA-co-TRIS-co-NVP) LbL coated hydrogel/CL (300 μm) Water uptake: 119%	44 μg/mg hydrogel for LbL MIP vs. 9 μg/mg hydrogel for LbL NIP IF = 4.88	Sustained release (23% release in the first 1 h followed by a sustained release of effective concentrations for 10 days; 20–25 μg/mg hydrogel cumulative release)	Material characterization In vitro release studies: methods –sink model, and the physiological flow model (microfluidic device); medium–130 mM NaCl (pH = 6.9), 36 °C In vitro antibacterial activity (S*. aureus* and *S. epidermidis*)	The combination of MI and LbL coating-sustained double release of moxifloxacin and diclofenac	[41]
Moxifloxacin and Diclofenac	Non-covalent imprinting; MAA/EGDMA/AIBN; 60 °C, 18 h	CI26Y hydrogel/IOL Water uptake: approx. 25%	Diclofenac: 16 μg/mg hydrogel for MIP vs. 12.2 μg/mg hydrogel for NIP IF = 1.31 Moxifloxacin: 28 μg/mg hydrogel for both MIP and NIP.	Diclofenac–sustained release up to 14 days (71% for MIP and 81% for NIP cumulative release) Moxifloxacin–sustained release for up to 14 days (83% for MIP and 81% for NIP cumulative release)	Material characterization In vitro release studies: method–sink model; medium–PBS (pH = 8), 36 °C Mathematical model for the prediction of the In vivo therapeutic efficacy In vitro irritability tests, cytotoxicity tests and antibacterial tests	MI Intraocular lens material for the simultaneous delivery of diclofenac and moxifloxacin	[42]
Minocycline	Non-covalent imprinting; AA/EGDMA/ 2,2’-Azobis(2,4-dimethyl-valeronitrile); 45 °C, 24 h	Poly-HEMA hydrogel/- Water uptake: 40% MIP vs. 35% NIP	0.243 μg/mg hydrogel for MIP vs. 0.082 μg/mg hydrogel for NIP IF = 2.9	Sustained release for 48 h (2.294 μg/mg hydrogel for MIP and 1.696 μg/mg hydrogel for NIP cumulative release)	Material characterization In vitro release studies: method–sink model; medium–NaCl 0.9%, 37 °C	Molecular dynamics simulations to select the suitable monomer and the optimum amount of crosslinker	[43]
Polymyxin B	Non-covalent imprinting; AA/EGDMA/AIBN; 50 °C, 12 h and 70 °C, 24 h	Poly-HEMA hydrogel/CL (400 μm) Water uptake: 37%	Approx. 90 mg/g disc	Sustained release for up to 14 days (41% drug released in 7 days; 40 mg/g disc cumulative release)	Material characterization In vitro release studies: method–sink model; medium–NaCl 0.9% In vitro antibacterial activity (*P. aeruginosa*)		[44]
Voriconazole	Non-covalent imprinting; AM, MMA/EGDMA/ benzoyl peroxide; 70 °C, 24 h	Collagen shield	Binding capacity: 82.79% for MIP vs. 20.65% for NIP IF = 4.01	In vitro release studies: Sustained release-73.5% after 24 h, and 85.39% after 48 h In vivo release studies: initial drug release up to Cmax = 62.3 ng/mL (24 h), for collagen-MIP vs. burst drug release up to Cmax = 47.38 ng/mL (3 h) followed by an exponential decrease for the voriconazole solution	Material characterization In vitro release studies: method-sink model; medium-artificial lacrimal fluid (32 °C) In vitro antifungal activity Ex vivo cornea permeation study In vivo studies on albino rabbits	In vivo release studies: 29.09 h MRT for collagen-MIP vs. 6.86 h MRT for voriconazole solution; sustained release with a 7.53-fold increase in bioavailability as compared to the voriconazole solution	[45]
Valacyclovir or Acyclovir	Non-covalent imprinting; MAA/EGDMA/AIBN; 50 °C, 12 h and 70 °C, 24 h	Poly-HEMA hydrogel/CL (450 μm) Water uptake: 90%	Valacyclovir: approx. 4.5 mg/g disc for MIP Acyclovir: approx. 2 mg/g disc for MIP	Valacyclovir: sustained release for 10 h (3.5–4 mg/g disc cumulative release) Acyclovir–0.6 mg/g disc accumulative release (irreversible binding of significant amount of template)	Material characterization In vitro eye compatibility test In vitro release studies: medium–artificial lacrimal fluid In vitro drug permeability studies (bovine and porcine cornea and sclera)	Computational modeling to elucidate drug–functional monomer interactions In vitro drug permeability studies–accumulation in the cornea and penetration through the sclera for valacyclovir	[46]
Diclofenac	Non-covalent imprinting; DEAEM/PEG200DMA/AIBN; UV, 8 min, 35 °C	Poly-HEMA hydrogel/CL (105 μm) wollen polymer volume fractions: 1.535	0.015–0.02 mg/mg hydrogel	Sink model: sustained release for 72 h Physiological flow model: Zero-order kinetics, linear release profile up to 48 h, (release rate 6.75 µg/h)	Material characterization In vitro release studies: methods– infinite sink model and physiological flow model (microfluidic device); medium–artificial lacrimal fluid (34 °C)	Zero-order release under physiological volumetric flow rates	[47]
Prednisolone acetate	Non-covalent imprinting; MAA/EGDMA/AIBN; 60 °C, 24 h	Poly-HEMA hydrogels/CL (400 μm) Water uptake: 62–63%	58 μg/disc for MIP vs. 39 μg/disc for NIP IF = 1.49	Sustained release for 48 h (64% drug in 48 h for MIP vs. 78% drug release within 8 h for NIP)	Material characterization In vitro release studies: methods–sink model, medium-0.9% NaCl and artificial lacrimal fluid (pH = 8), 37 °C		[48]
HPMC, trehalose, ibuprofen, prednisolone	Non-covalent imprinting; AA, 4VPh/EGDMA, PEG200DMA/Darocur 1173; UV, 1.5 min., room temperature	Poly (TRIS-co-DMA-co-DMS-R11)/CL (100 μm) Water uptake: not reported	HPMC and trehalose: 500 μg/lens Ibuprofen and prednisolone: 150 μg/lens	Sustained release HPMC: 20% drug release in 7 days (release rate 14–18 μg/day), and complete release in 36 days Trehalose: 90% drug release/24 h (linear release, 17 μg/h) Ibuprofen and prednisolone: 30% drug release in 1 day (release rate 45 μg/day)	Material characterization In vitro release studies: methods–sink model, and physiological flow model (microfluidic device); medium–water, NaCl 0.9%, NaCl 5%, PBS (pH = 7.4), artificial lacrimal fluid (pH = 8), 34 °C	Simultaneous load and release of HPMC, trehalose, ibuprofen and prednisolone	[49]
Fluorometholone	Non-covalent imprinting; MAA/EGDMA/AIBN; UV, 40 min, room temperature	Poly-HEMA hydrogels/CL (200 μm) Water uptake: 53.75–57.25%	61 μg/lens for MIP vs. 40 μg/lens for NIP IF = 1.53	Sustained release-59.7% and 92.8% of drug was released from MIP and NIP, respectively, within the first 8 h	Material characterization In vitro release studies: method–sink model; medium-0.9% NaCl (pH = 5.5), artificial lacrimal fluid (pH = 8), 37 °C		[50]
Bimatoprost	Non-covalent imprinting; MAA/EGDMA/Irgacure; UV, 15 min, room temperature	Poly (HEMA-co-DMA-co-siloxane) hydrogel/CL (thickness not reported) Water uptake: 88% MIP vs. 92% NIP	14.76 μg/lens for MIP vs. 10.75 μg/lens for NIP IF = 1.37	In vivo release study: sustained release up to 36–60 h-C_max_ (5 min)-56.26 μg/mL for MIP vs. 62.35 μg/mL for NIP and 145.26 μg/mL for 0.03% bimatoprost eye drop solution	Material characterization In vitro release studies: method-sink model; medium–artificial lacrimal fluid (pH = 8), 34 °C In vivo release studies: white New Zealand rabbits	The *in-vivo* study-low burst release and increase in bimatoprost MRT for MIP-SCL vs. conventional soaked SCL (NIP) and 0.03% bimatoprost eye drop solution	[51]
Brimonidine	Non-covalent imprinting; MAA/EGDMA/AIBN; 50 °C, 24 h	Poly-HEMA hydrogel/CL (400 μm) Water uptake: 47%	3 μg/mg hydrogel for MIP vs. 1.75 μg/mg hydrogel for NIP IF = 1.71	Sustained release for 48 h(39-50% drug released from MIPs vs. 70% drug released from NIP in the first 30 min)	Material characterization In vitro release studies: medium–NaCl 0.9%, and artificial lacrimal fluid, pH = 8, 37 °C		[52]
Acetazolamide	Pivot based imprinting; Zn(II) methacrylate, 4VI, HEAA/EGDMA/AIBN; 50 °C, 12 h and 70 °C, 24 h	Poly-HEMA hydrogel/CL (900 μm) Water uptake: 80–85%	Acetazolamide: 3.28 mg/g hydrogel for MIP vs. 3.15 mg/g for NIP Ethoxzolamide: 1.71 mg/g hydrogel for MIP vs. 1.55 mg/g for NIP	Sustained release for 2 weeks (Acetazolamide: 20% released in 6 h and 50% released in 192 h; Ethozxolamide: 15% released in 6 h and 25% released in 192 h)	Material characterization In vitro release studies: method–sink model; medium–0.9% NaCl, room temperature Cytocompatibility tests (Balb/3T3 Clone A31 cell line)	Receptor-inspired hydrogel Highest affinity and better control of release for the non-imprinted networks; the use of functional monomers that best mimic carbonic anhydrase receptors is critical	[53]
Acetazolamide	Pivot based imprinting; Zn(II) nitrate hexahydrate, 4VI, HEAA/Zn(II)/EGDMA/AIBN; 50 °C, 12 h and 70 °C, 24 h	Poly (DMA-co-NVP) hydrogel/CL (900 μm) Water uptake: 80–85%	Acetazolamide: 4.11 mg/g hydrogel for MIP vs. 3.81 mg/g hydrogel for NIP Ethoxzolamide: 1.34 mg/g hydrogel for MIP vs. 1.47 mg/g hydrogel for NIP	Sustained release for 9–12 h (Acetazolamide: 50% drug release in the first hour; Ethoxzolamide: 50–70% drug released in 12 h)	Material characterization In vitro release studies: method–sink model; medium–0.9% NaCl, room temperature Cytocompatibility tests (Balb/3T3 Clone A31 cell line)	Receptor-inspired hydrogel The use of functional monomers that best mimic carbonic anhydrase receptors is critical	[54]
Dorzolamide	Non-covalent imprinting; MAA/EGDMA/AIBN; 50 °C, 24 h	Poly-HEMA hydrogel/CL (400 μm) Water uptake: approx. 47%	105 μg/disc for MIP vs. 77 μg/disc for NIP IF = 1.36	Sustained drug release for 48 h (24% drug released from MIP vs. 62% drug released from NIP in the first 0.5 h, and 51% drug released from MIP vs. 80% drug released from NIP in 3 h)	Material characterization In vitro release studies: method –sink model; medium–0.9% NaCl and artificial lacrimal fluid (pH = 8), 37 °C		[55]
Atorvastatin	Non-covalent imprinting; EGPEM, AEMA, APMA/EGDMA/AIBN; 50 °C, 12 h and 70 °C, 24 h	Poly-HEMA hydrogel/CL (300 μm) Water content: 90%	6.69 mg/g hydrogel for MIP vs. 5.81 mg/g hydrogel for NIP IF = 1.15	Sustained release for 7 days; (~7 mg/g for MIP vs. ~6 mg/g for NIP after 1 week)	Material characterization In vitro release studies: method–sink model; medium-artificial lacrimal fluid (pH = 7.4), 37 °C Ocular irritancy test (hen’s egg test chorioallantoic membrane) Cytocompatibility tests (Balb/3T3 fibroblasts cells) Ex vivo cornea and sclera permeability and accumulation of atorvastatin	Computational modeling docking to elucidate drug–functional monomer interactions	[56]
Hyaluronic acid	Non-covalent imprinting; AM, NVP, DEAEM/Nelfilcon A/Irgacure 2959; UV, 45 s	PVA-based commercial formulation/CL (127 μm) Swollen polymer volume fractions: 0.23–0.29	Not reported	Sustained release for 48 h (release rate: 12 μg/h in the first 6–10 h, and nearly liner release profile at a rate of 4 μg/h in the next 10 to 30 h)	Material characterization In vitro release studies: method–sink method, medium-artificial lacrimal solution (pH = 8), 35 °C	Receptor-inspired hydrogel (microdomains that resemble the binding sites in the cell-surface glycoprotein CD44) Hyaluronic acid diffusion coefficients and release profiles can be significantly altered by the changes in mass content and by the relative proportions of functional monomers	[57]
HPMC	Non-covalent imprinting; AA/PEG200DMA, EGDMA/Darocur 1173; UV, 1.5 min	Lotrafilcon B, commercial silicone hydrogel lens (Silicon hydrogel macromer, TRIS and DMA)/CL (350 μm) Water content: 33%	Not reported	Sustained release - 1000 μg of HPMC over a period of up to 60 days in a constant manner at a rate of 16 μg/day	Material characterization In vitro release studies: method–sink method, medium-deionized water (pH = 6.4), 34 °C		[58]

AA: acrylic acid; ACN: acetonitrile; AEMA: 2-aminoethyl methacrylate hydrochloride; AIBN: 2,2’-azo-bis-iso-butyronitrile; APMA: N-(3-aminopropyl) methacrylamide hydrochloride; CI26Y: 80–90% HEMA, 10-20% MMA, <1% EGDMA, <1% 2-(4-benzoyl-3-hydroxyphenoxy) ethyl acrylate); CL: soft contact lens; Darocur 1173: 2-Hydroxy-2-methylpropiophenone; DEAA: N,N-diethylacrylamide; DEAEM: diethylaminoethyl methacrylate; DMA: dimethyl acrylamide; DMS-R11: methacryloxypropyl terminated polydimethylsiloxane; DMPAP: 2,2-dimethoxy-2-phenylacetophenone; EGDMA: ethylene glycol dimethacrylate; EGPEM: ethylene glycol phenyl ether methacrylate; HA: hyaluronic acid; HEAA: N-hydroxyethyl acrylamide; HEMA: 2-hydroxyethyl methacrylate; HPMC: hydroxypropyl methylcellulose; IF: imprinting factor; Irgacure: 2-Hydroxy-1-{4-[4-(2-hydroxy-2-methyl-propionyl)-benzyl]-phenyl}-2-methyl-propan-1-one; IOL: intraocular lens; LbL: layer-by-layer; MAA: methacrylic acid; MMA: methyl methacrylate; MRT: mean residence time; NVP: N-vinyl pyrrolidone; PEG200DMA: polyethylene glycol 200 dimethacrylate; PBS: phosphate buffered saline; TEGDMA: tetraethylene glycol dimethacrylate; TRIS: methacryloxypropyl-tris-(trimethylsiloxy) silane; 4VI: 4-vinylimidazole; 4VP: 4-vinyl pyridine; 4VPh: 4-vinyl phenol.

## Data Availability

Not applicable.

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
