# Peer review of "Perspectives of Molecularly Imprinted Polymer-Based Drug Delivery Systems in Ocular Therapy"

_polymers, 2021, doi:10.3390/polym13213649_

Round 1

Reviewer 1 Report

The manuscript is well written and represents an initial trial to use commercial polymers for ocular therapy and contains data that is worth of publication and criticism. 

Author Response

We would like to thank Reviewer 1 for acknowledging the relevance of our review manuscript. During the revision process the manuscript has undergone additional improvements in terms of style and spell check ("track changes" has been used in the revised form).

Reviewer 2 Report

Perspectives of molecularly imprinted polymer-based drug delivery systems in ocular theraphy

It is possible to publish this manuscript in Polymers Journal with very minor corrections.

-line 103, RPE

Need the original spelling before using the abbreviation. change it with line 213.

-Line 194, PLGA

Need original spelling before using abbreviation.

Some opinions as a reviewer

-The cited papers used timolol maleate ( or ketotifen fumarate) to imprint into soft contact lens (CL) containing acrylic acid ( or methacrylic acid) rather than pure timolol only. Timolol maleate can be neutralized (?) or complexed already between N of timolol molecule and maleic acid. In the re-adsorption process of timolol to CL, the sorption can be more difficult or more predominant at one component between timolol and maleate. It is very tedious and hard work to read many papers. But it is very thankful to authors if they may propose the opinions. The published papers are on open discussion. Usually main component of soft CL is HEMA without –COOH groups which is used only for complexation with active pharmaceutical ingredient. Thus, if possible, the mal-function of –COOH groups to eye can be considered even though this is a review paper.

-An advantage on DDS of MIP is long time releasing than NIP. Considering the dropping frequencies of the eye drops to naked eye, the releasing time of NIP can be compared if possible. The releasing time of NIP can be longer than naked eye dropping. As the experts in the field of ocular therapy and pharmacy, such points should be point out for the next researchers.  

And also the real experimental figure comparing the releasing time should be cited for readers to understand easily.
